# Co-chaperone involvement in knob biogenesis implicates host-derived chaperones in malaria virulence

**Mathias Diehl**[1], **Lena Roling**[2], **Lukas Rohland**[3], **Sebastian Weber**[4], **Marek Cyrklaff**[1], **Cecilia P. Sanchez**[1], **Carlo A. Beretta**[5,6], **Caroline S. Simon**[1], **Julien Guizetti**[1], **Julia Hahn**[2], **Norma Schulz**[2], **Matthias P. Mayer**[3], **Jude M. Przyborski**[2]*

**1** Parasitology, Centre for Infectious Diseases, Heidelberg, Germany, **2** Biochemistry and Molecular Biology, Justus Liebig University, Gießen, Germany, **3** Center for Molecular Biology (ZMBH), Heidelberg, Germany, **4** Electron Microscopy Core Facility, Heidelberg University, Heidelberg, Germany, **5** Nikon Imaging Center, Heidelberg University, Heidelberg, Germany, **6** CellNetworks Math-Clinic at Heidelberg University, Heidelberg, Germany

* jude.przyborski@ernaehrung.uni-giessen.de

**Data Availability Statement:** All relevant data are within the manuscript and its Supporting information files.

## Abstract

The pathology associated with malaria infection is largely due to the ability of infected human RBCs to adhere to a number of receptors on endothelial cells within tissues and organs. This phenomenon is driven by the export of parasite-encoded proteins to the host cell, the exact function of many of which is still unknown. Here we inactivate the function of one of these exported proteins, PFA66, a member of the J-domain protein family. Although parasites lacking this protein were still able to grow in cell culture, we observed severe defects in normal host cell modification, including aberrant morphology of surface knobs, disrupted presentation of the cytoadherence molecule PfEMP1, and a total lack of cytoadherence, despite the presence of the knob associated protein KAHRP. Complementation assays demonstrate that an intact J-domain is required for recovery to a wild-type phenotype and suggest that PFA66 functions in concert with a HSP70 to carry out host cell modification. Strikingly, this HSP70 is likely to be of host origin. ATPase assays on recombinant protein verify a functional interaction between PFA66 and residual host cell HSP70. Taken together, our data reveal a role for PFA66 in host cell modification, strongly implicate human HSP70s as being essential in this process and uncover a new KAHRP-independent molecular factor required for correct knob biogenesis.

## Author summary

To survive in the human body, the malaria parasite invades and lives within human red blood cells. Once within the red blood cell, the parasite renovates the host cell to its own needs. Here we have studied which factors from both parasite and host cell are required for this renovation process, and discover that human chaperone proteins, referred to as HSP70, are required. It appears that a particular parasite-derived protein, PFA66, recruits and modifies the function of the human HSP70. As this interaction between a parasite and

**Funding:** This work was funded by the Deutsche Forschungsgemeinschaft (DFG, German Research foundation, DFG.de) grant 351262938 and 391524768 to JMP and 462625623 (MA1278/10-1) to MPM. The funders had no role in study design, data collection and analysis, decision to publish, or preparation of the manuscript.

**Competing interests:** The authors have declared that no competing interests exist.

**Abbreviations:** (i)RBC, (infected) red blood cell; (r)STED, (rescue) stimulated emission depletion; BSD, blasticidin; CSA, chondroitin-sulphate-A; EQT, equinatoxin; IFA, immunofluorescence assay; KAHRP, knob-associated histidine-rich protein; NPP, new permeability pathway; PPM, parasite plasma membrane; PV(M), parasitophorous vacuole (membrane); RBCM, RBC membrane; SEM, scanning electron microscopy; SLI-TGD, selection-linked integration targeted gene disruption; TEM, transmission electron microscopy.

human protein is novel and essential for parasite survival, our study identifies a potential Achilles' Heel which may be targeted for development of new anti-malaria therapies.

## Introduction

*Plasmodium falciparum* causes the most severe form of malaria in humans, *malaria tropica*, responsible for over 200 million clinical cases and 400,000 deaths *per annum*, mainly in children under the age of 5 and mostly in sub-Saharan Africa [1]. The pathology associated with malaria infection is largely due to the ability of infected human RBCs (red blood cells, RBC) to adhere to a number of receptors on endothelial cells within tissues and organs [2]. This cytoadherence results in reduced blood flow in the affected areas, hypoxia and (in cerebral malaria) increased intracranial pressure [2,3]. The phenomenon of cytoadherence results from parasite-induced host cell modification in which parasite-encoded proteins are transported to and exposed at the surface of the infected host cell, where they mediate endothelial binding and antigenic variation [4–6]. In addition to these surface proteins, parasites also encode, express, and export a large number of other proteins to the infected RBC [7–9,10,11]. Many of these proteins are specific to *P. falciparum*, and their function is still not well understood, partly due to limitations in reverse genetic systems [9,12,13].

Within the predicted 'exportome' are 19 proteins belonging to the family of J-domain proteins (JDPs, also called HSP40s), and this exported family appears to be expanded in the Laveranian clade, suggesting important functions in these parasite species [9,14]. In other systems, HSP40s act as cochaperones for HSP70, a protein family that lies at the heart of proteostasis and other essential cellular processes [15]. Previous studies have localised several exported members of the *Pf*HSP40 family to various structures within the infected RBC, including RBC plasma membrane, Maurer's clefts, knobs, and J-dots [13,16,17]. Knockout studies suggest that, although some of the exported HSP40s are essential, others can be deleted without any observable phenotype, and knockout of others results in aberrant cellular morphology, cytoadherence, and rigidity of infected RBC (iRBC), suggesting a potential role for this protein family in host cell modification and virulence characteristics [12,13] (S1 Table). One exported HSP40, PFA66 (encoded by *PF3D7_0113700*, formerly *PFA0660w*), was previously localised to J-dots, novel structures within the iRBC containing further exported parasite-encoded HSP40s, an exported parasite-encoded HSP70 (*Pf*HSP70-X), and a number of other exported proteins [16,18,19]. An earlier medium throughput knockout study failed to generate parasites deficient in PFA66, and therefore its function in the parasite's lifecycle remains elusive [12].

In this study, we utilise selection-linked integration-targeted gene disruption (SLI-TGD [20]) to generate parasites expressing a non-functional PFA66 truncation mutant and characterise the resulting parasite lines. We find that inactivation of PFA66 function leads to dramatic aberrations in host cell modification, especially in knob morphology, capacity for cytoadherence and surface exposure of the virulence factor PfEMP1. Our data suggest an important role for exported HSP40s in parasite pathogenicity. Additionally, our data strongly implicate residual human HSP70 in parasite-induced host cell modification.

## Results

### Generation of PFA66 truncation and complementation cell lines

Genetic manipulation via single crossover was performed using a selection-linked integration targeted gene disruption strategy [20]. Integration of the plasmid would lead to the production

of a GFP-tagged, truncated, non-functional PFA66 protein lacking the entire C-terminal substrate binding domain (SBD), referred to as dPFA. We transfected CS2 parasites that had previously been freshly selected for binding to chondroitin-sulphate-A (CSA) as this binding phenotype would be essential for later characterisation [21,22]. Plasmid integration into the *PFA0660w/PF3D7_0113700* locus was verified via PCR using primers designed to yield products upon integration of the plasmid into the genome via primers spanning the integration site (Fig 1A). Appearance of bands representing the 5' and 3' integration, as well as the disappearance of bands representing the endogenous *PFA0660w* locus, demonstrated specific integration of the plasmid into the genome (Fig 1B and S1A Fig), yielding parasite line CS2Δ*PFA* (referred to as Δ*PFA*). Immunodetection using an α-GFP antibody revealed the presence of dPFA::GFP fusion at the predicted size in cell lysates derived from Δ*PFA* but not parental CS2 parasites (predicted MW 39 kDa, Fig 1C). Additionally, one further higher molecular weight band was observed, likely representing a non-skipped fusion (predicted MW 68 kDa, Fig 1C[+]), similar to that previously reported [23]. Although levels of the control SERP protein were similar in all samples analysed, verifying equal sample loading, the dPFA::GFP fusion appears to be present at a lower level than in parasites expressing endogenous full length PFA tagged with GFP (Fig 1C). Although we are unsure or the reasons for this, the higher abundance of the commonly observed GFP degradation product around 25kDa (Fig 1C[*]) may suggest lower protein stability. Together, this data indicated successful integration of the vector into the *PFA0660w* locus, leading to the production of a truncated product d*PFA* lacking the substrate binding domain required for correct localisation and function [24]. To verify that any aberrant phenotypes observed were due to a lack of PFA66 and not second site events, we generated a complementation line that expressed a full-length, functional 3xHA (hemagglutinin)-tagged copy of PFA66 from an episome, under the control of the endogenous PFA66 promoter, referred to as Δ*PFA*[PFA::HA]. Expression of this complementation construct was verified by immunoblotting using α-HA antiblodies (Fig 1D).

## dPFA is soluble within the host cell cytosol

Full-length PFA66 has previously been localised to the J-dots, and a follow up study suggested that the SBD of exported HSP40s is required for correct localisation [16,24]. Furthermore, the SBD of PFA66 has previously been shown to bind cholesterol, and this is also likely to be required for correct localisation and thus protein function [25]. As dPFA lacks the SBD, but still contains both an N-terminal signal peptide and a recessed PEXEL trafficking signal, we expect export of dPFA::GFP to the host cell [7,8]. Live cell imaging and IFA failed to detect dPFA. This is likely due to the low abundance, and dispersion of the truncated protein throughout the entire host cell. We therefore used equinatoxin (EQT) to selectively permeabilise the RBC plasma membrane and allow sub-cellular fractionation of iRBC [26,27]. Immunodetection using antibodies against the compartment-specific markers SERP (parasitophorous vacuole), human HSP70 (HsHSP70, RBC cytosol), aldolase (ALDO, parasite cytoplasm), and GFP reveals co-fractionation of dPFA with HsHSP70, verifying that dPFA::GFP is found in the host cell cytosol and furthermore that dPFA::GFP is found in the soluble phase and not in the membrane fraction as we have previously demonstrated for the full-length protein (S1B Fig [16]). Taken together, these data suggests that deletion of the SBD of PFA66 leads to a non-functional protein.

## Truncation of PFA66 affects novel permeability pathway (NPP) activity and confers a small growth advantage

Exported parasite proteins carry out a multitude of functions supporting the survival of *P. falciparum* parasites. One of these is the establishment of NPPs of the iRBC to support the uptake

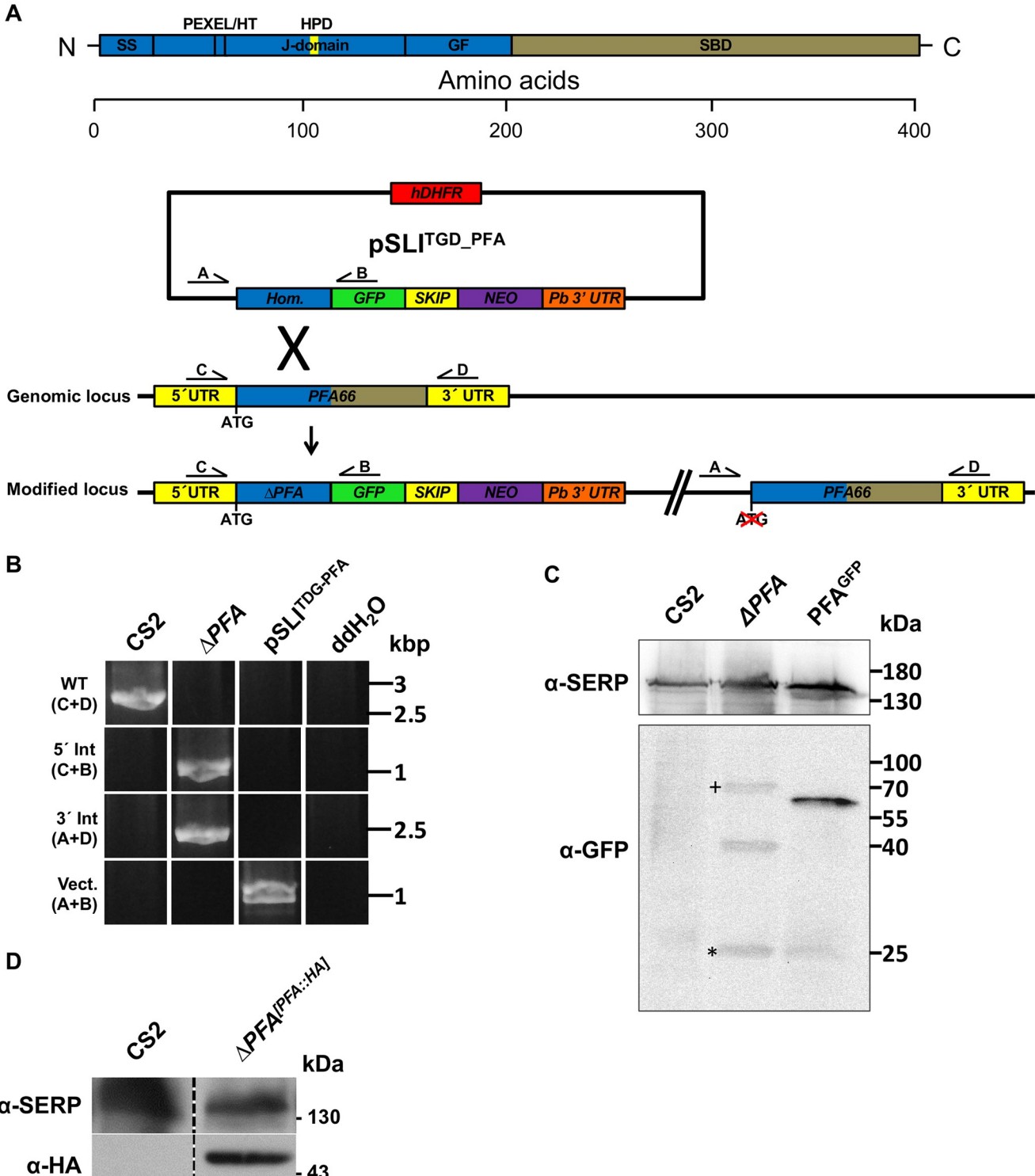

**Fig 1. Generation and verification of the Δ*PFA* cell line.** A) Strategy for inactivation of *PFA0660w* via selection-linked integration. Expression of a neomycin resistance marker (NEO) is coupled to integration of the plasmid pSLI^TGD_PFA into the genomic *PFA0660w* locus, leading to expression of a truncated (likely inactive) PFA66 missing its substrate binding domain (SBD). Production of PFA and NEO as separate proteins is mediated with a SKIP peptide. B) Integration PCR using gDNA extracts from the cell line Δ*PFA* and the parental cell line CS2 verifies integration of the plasmid pSLI^TGD_PFA into the *PFA0660w* gene in the cell line Δ*PFA*. Amplification of the wild type *PFA0660w* locus with the primers C and D is only successful in the parental strain CS2 since integration of the plasmid dramatically increases product size. PCRs using primers spanning the junctions of the integration sites C and B for the

5' region and A and D for the 3' region) demonstrate disruption of *PFA0660w*. C) Western blot verifies truncation of *PFA0660w* in Δ*PFA*. The truncated fusion protein was detected using an α-GFP antibody, while the parasite protein SERP served as a loading control. <sup>+</sup>unskipped product, *GFP degradation product. D) Immunodetection verifies expression of HA-tagged PFA66 in the complementation cell line Δ*PFA*[*PFA::HA*].

of essential nutrients [28,29]. To investigate a potential role of PFA66 in NPP activity, we used a sorbitol uptake/lysis assay, which revealed that RBC infected with Δ*PFA* parasites show a small but significantly reduced sorbitol-induced lysis, implying a reduced NPP capacity (S1C Fig). To examine whether this reduction in NPP activity affects parasite viability, growth of both cell lines was compared over four growth cycles (~8 days) using flow cytometry. Surprisingly, a slight but significant growth advantage (calculated as below 1% advantage per cycle) of the truncation cell line compared to the parental cell line was observed (S1D Fig).

## Truncation of PFA66 causes severely deformed knob morphology

Knobs are electron-dense protrusions of the iRBC surface that help correctly present the major virulence factor *Pf*EMP1, thus facilitating iRBC cytoadhesion and concomitantly increasing clinical pathology [4–6,30,31]. As exported HSP40s have previously been implicated in knob formation [12,17], we used scanning electron microscopy to visualise knobs on the surface of CS2- and Δ*PFA*-iRBCs. RBCs infected with the parental parasite line CS2 showed normal knob morphology, with an even distribution of small knobs over the entire surface of the infected cell (Fig 2A and S2 Fig). Small variations in knob number and size are observed and are likely due to slightly different developmental stages of the parasite (compare Fig 2B and 2C to S6A, S6B, S6D and S6E Fig). In stark contrast, RBCs containing Δ*PFA* displayed, in addition to a population of normal knobs, knobs with extremely aberrant morphology. These knobs varied in their aberration, and included vastly extended-, wide tall-, wide flat-, and branched- knobs (S2 Fig). Some of the elongated knobs reached lengths of ~0.7 μm. To distinguish these abnormal structures from classical knobs, we will refer to them as "extended knobs" (eKnobs). For purposes of quantification, we manually counted and classified by sight knobs/eKnobs to one of three classes I) normal/small knobs II) abnormal/enlarged eKnobs III) deformed/elongated eKnobs. This analysis revealed a significant reduction in the overall number of knobs/eKnobs in Δ*PFA*-infected RBCs when compared to wild type CS2 (CS vs Δ*PFA*, Fig 2B). 22% of surface structures exhibited abnormal morphology in Δ*PFA*-infected RBCs (Fig 2C). Interestingly, we occasionally observed extended knob-like structures on the surface of RBCs infected with CS2 parasites (Fig 2C) at a level of 2%. As a control, we complemented Δ*PFA* function with full-length PFA66 expressed from an episome (Δ*PFA*[*PFA::HA*]), and both density and morphology of eKnobs/knobs returned to wild type levels (Fig 2B and 2C).

Transmission electron microscopy (TEM) on thin sections prepared from RBCs infected with either WT CS2 or Δ*PFA* parasites substantiated our observations. CS2 parasites produced clearly defined electron-dense knob structures of a restricted diameter and height above the RBC plasma membrane, whereas Δ*PFA* displayed eKnobs extending from the RBC surface into the external medium (Fig 2D and S3 Fig). The lumen of these eKnobs was often extremely electron dense, hinting at their molecular relation to knobs. Occasionally we observed membrane-bound structures extending from RBCs infected with CS2 parasites, but the lumen of these structures was not electron dense; therefore, these structures cannot be classed as eKnobs. The morphology of Maurer's clefts showed the normal variation in morphology in both samples (S3 Fig). Several clefts appear to lie perpendicular to the host cell membrane, but this phenotype was observed in both samples (S3 Fig).

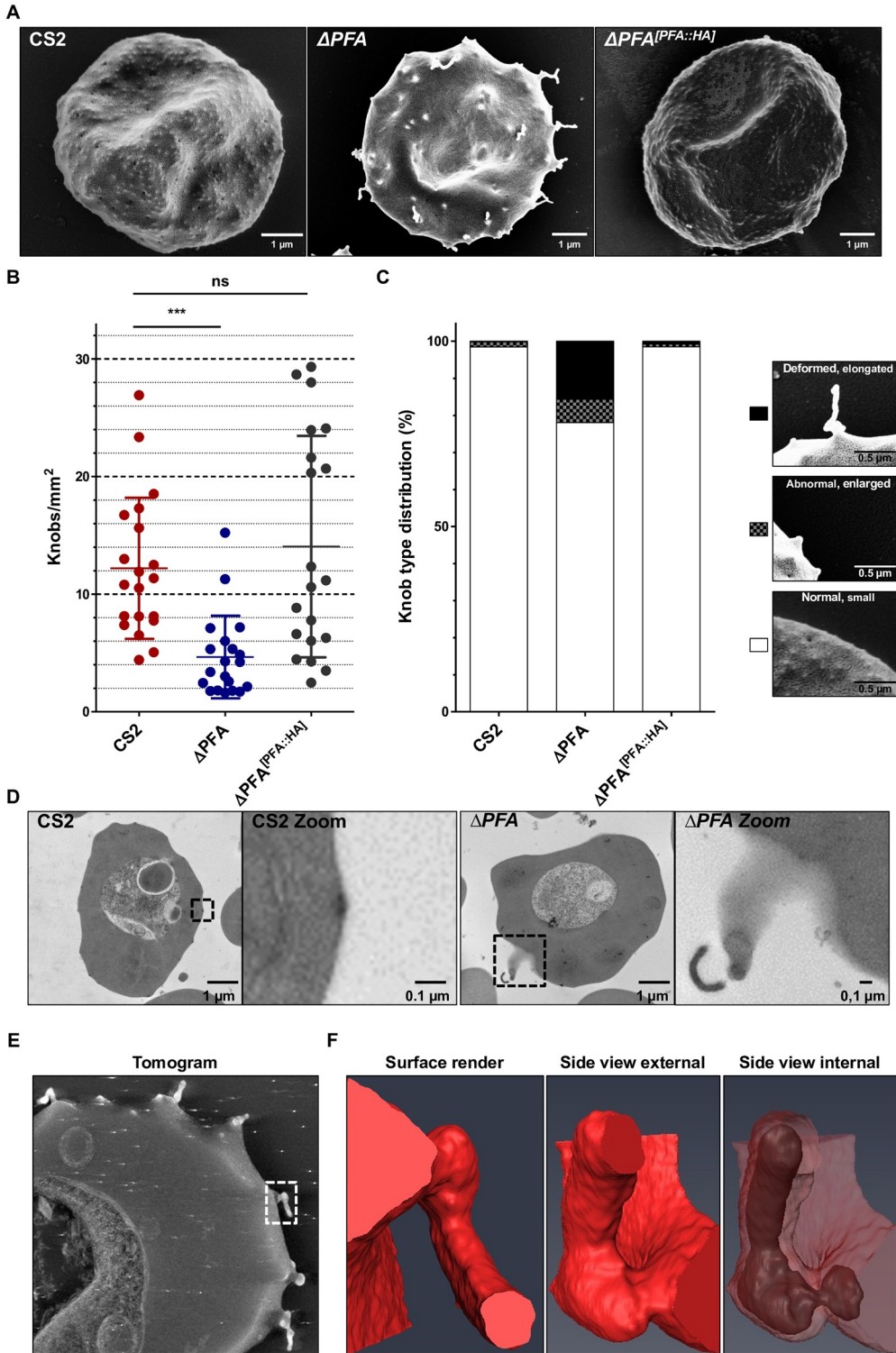

**Fig 2. Electron microscopy reveals deformed knob morphologies of Δ*PFA* iRBCs.** A) Scanning electron microscopy shows knobs on the surface of iRBCs. The mutant phenotype of Δ*PFA* is alleviated upon reintroduction of episomally expressed PFA66 in Δ*PFA*[PFA::HA]. More pictures can be found in S2 Fig. B) Quantification of knob density via ImageJ in SEM pictures (n = 20) shows significantly fewer knobs on Δ*PFA* iRBCs. Knob density is restored in the complementation cell line CS2Δ*PFA*[PFA::HA]. C) Quantification of knob morphology across all iRBCs. Knobs were grouped into three categories: small knobs, enlarged knobs, and elongated knobs. Then every knob on 20 SEM pictures

of the three strains was assigned to one of these categories. Each bar represents the distribution of these knobs in the three categories across all pictures of a strain. The ΔPFA strains display an increase in deformed and enlarged knob morphologies compared to CS2 and ΔPFA$^{[PFA::HA]}$. D) Internal view of the deformed knobs/eKnob via transmission electron microscopy of thin slices. More pictures can be found in S3E Fig) Electron tomography reveals electron-dense material at the base and interior of deformed knobs/eKnobs. The marked area denotes the structure shown in F. F) 3D segmentation of discrete densities within a deformed knob/eKnob depicted with electron tomography. The example shows a severely deformed knob/eKnob. Additionally, electron dense material was detected at the base and inside of these structures.

## Electron-dense material is a scaffold for eKnob structure

Intrigued by the apparent electron-dense core of eKnobs in TEM, we conducted electron tomography analysis on thick sections prepared from ΔPFA-iRBCs. Subsequent 3D reconstruction and surface rendering of the distinct densities in tomographic volumes allows a high-fidelity glimpse into the fine structure of eKnobs. This analysis verified the presence of electron-dense material within and at the base of eKnobs (Fig 2E). This material fills out the entire eKnob rather than only lining the structure, and the distribution of this material closely matched the structure of the eKnob. In some cases (3 out of 10), the electron-dense material at the base of the eKnob was connected via a thin bridge to the material inside of the eKnob (Fig 2F, S1 Video).

## KAHRP distribution is changed in ΔPFA-iRBCs

KAHRP has long been held to be a crucial knob-associated protein as parasites lacking this protein no longer form knobs [30,31], and KAHRP truncations show varying aberrant knob phenotypes [32]. For this reason, we investigated the localisation of KAHRP in RBCs infected with our ΔPFA cell line. IFA on fixed cells demonstrated a punctate distribution of KAHRP in cells infected with both wild type and mutant cell lines (Fig 3A). Automated analysis with a self-generated ImageJ plugin paired with Ilastik [33] revealed that structures labelled for KAHRP were equally numerous but statistically larger in diameter in RBCs infected with ΔPFA than with CS2 (Fig 3B and 3C). Localisation of other exported parasite proteins via IFA showed no dramatic difference between the cell lines (S4 Fig). To exclude that the KAHRP result was due to non-specific binding of antibodies, for example, we episomally expressed a KAHRP::mCherry fusion in both WT and ΔPFA parasites (S5A Fig). Surprisingly, considering the relatively low resolution of live cell epifluorescence microscopy, (but preserving membrane integrity and cell morphology), structures labelled with KAHRP::mCherry could be seen to emerge from the surface of RBCs infected with ΔPFA, possibly representing eKnobs (Fig 3D, S3–S5 Videos). Based on the previous result, we wanted to understand whether KAHRP is directly associated with eKnobs and used immunogold labelling of thin sections derived from CS2 and ΔPFA iRBCs to localise KAHRP. Although we encountered high background staining of the RBC cytosol in both cases, analysis revealed considerably more label was associated with knobs/eKnobs than with the cytosolic background (Fig 3E and 3F). Analysis of the location of the KAHRP label in relation to the length of the knob/eKnob or the closeness to the RBC membrane (RBCM) showed no difference between WT and mutant, or between knob/eKnob morphologies (S5B and S5C Fig). To gain more insight into the nature of the structures labelled with α-KAHRP antibodies, we paired membrane shearing with immunolabelling and STED (stimulated emission depletion [34]) microscopy to study the nature of the structures labelled by α-KAHRP antibodies observed above. RBCs infected with CS2 or ΔPFA were allowed to bind to a cover slip, hypotonically lysed to obtain access to the internal leaflet of the RBC plasma membrane, fixed, immunodecorated using an α-KAHRP antibody, and imaged STED microscopy. This technique allows super-resolution visualisation of KAHRP structures

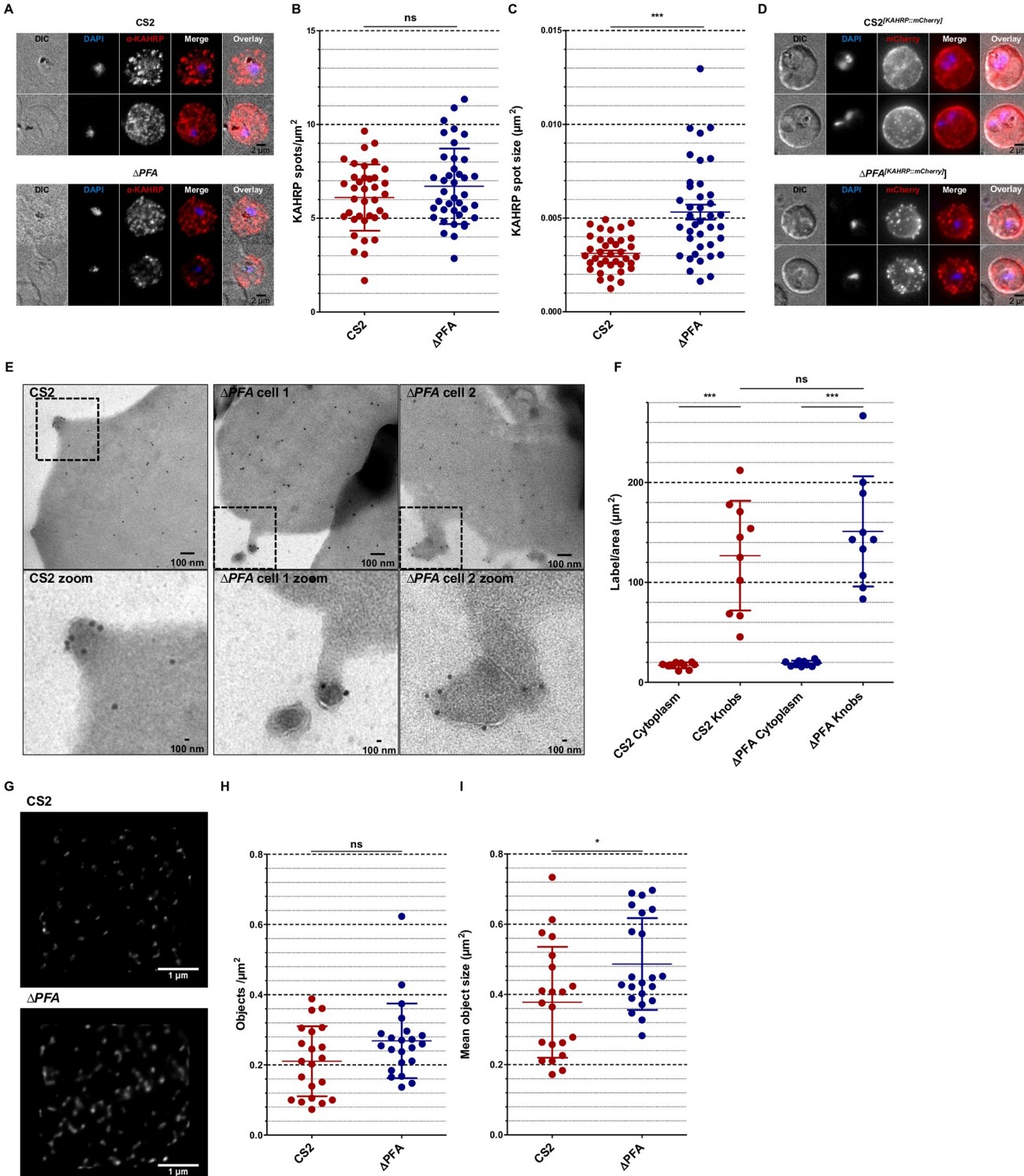

**Fig 3. ΔPFA iRBCs display altered KAHRP distribution.** A) IFA assay of MeOH-Ac-fixed CS2 and ΔPFA using α-KAHRP antibodies reveals punctate patterns. A trend was noticed towards bigger spots in the truncation strain and verified using automated measuring via an ImageJ algorithm (See Fig 3 B, C). D) Live cell imaging of DAPI stained CS2[KAHRP::mCherry] and ΔPFA[KAHRP::mCherry]. KAHRP::mCherry can be seen in both cell lines as punctate patterns; however, CS2 displays smaller and more dots. E) Immunogold labelling of iRBC sections in TEM using α-KAHRP antibodies. Images demonstrate label associated with normal knobs and deformed knobs in CS2 and ΔPFA, respectively. Framed areas can be seen enlarged below. F) Analysis of label density associated with the cytoplasm and area surrounding knobs. Label density is significantly higher in the area surrounding knobs than the cytoplasm for both

strains. G) STED imaging of the KAHRP associated with the internal RBC cytoskeleton. For this analysis CS2 and Δ*PFA* iRBCs were bound to a dish and then lysed hypotonically. The cell body was then washed away, and the remaining cytoskeleton remained as it would be seen from the inside of the iRBC. These samples were then interrogated with an α-KAHRP antibody and STED imaging. G) Representative images of the KAHRP patterns observed in STED from the CS2 and Δ*PFA* cell line. KAHRP signals were often found to be bigger in the truncation cell line. H) Computational analysis of KAHRP signals through a self-made ImageJ tool revealed no difference in KAHRP spot numbers between both cell lines. I) Investigation of mean object size demonstrated a slight increase of KAHRP spot size in Δ*PFA*.

from the luminal side of the RBC plasma membrane and can be used to monitor the assembly of KAHRP into knobs (and in this case, eKnobs). This analysis revealed a punctuate distribution of KAHRP beneath the membrane of RBCs infected with both wild type and mutant Δ*PFA* cells. Although the data initially suggests that there are more KAHRP structures in membranes from Δ*PFA*, automated counting and measurement shows that this impression is likely to be imparted due to a higher mean size of KAHRP objects rather than an actual increase in the absolute number of objects (Fig 3G, 3H and 3I).

## eKnobs contain ring/tube-like KAHRP structures but only small amounts of actin

To further investigate the relation of KAHRP with eKnob we used rSTED (rescue-stimulated emission depletion [35]) to image both RBCs infected with CS2 or Δ*PFA* parasites episomally expressing KAHRP::mCherry as above (S5A Fig). Infected cells were treated with RFP booster to amplify the mCherry signal, fluorescently labelled wheat germ agglutinin (WGA) to label the RBC glycocalyx (delineating the RBC membrane, RBCM), and phalloidin to stain host cell actin (Fig 4).

Line scan analysis reveals that eKnobs are bounded by the RBCM. In most cases, KAHRP is found between the RBCM and the actin cytoskeleton, and the KAHRP::mCherry signal also extends into the central cavity of the eKnobs. Distribution of host actin closely follows that of WGA, apart from at the base of eKnobs where it follows a path below the KAHRP staining. Fortuitous sectioning of several membrane-bounded eKnob*s* extending from the RBC reveals a ring of KAHRP::mCherry staining lining the luminal face of the eKnob. Phalloidin staining was absent in this case, although it was associated with a shorter eKnob (Fig 4, see S5D Fig for CS2 cell line).

## Chelation of membrane cholesterol but not actin depolymerisation or glycocalyx degradation causes partial reversion of the mutant phenotype in Δ*PFA*

A number of lipid- or protein-dependent mechanisms can induce the initiation of curvature of biological membranes and stabilisation of the resulting structures [36]. The eKnobs we observe here extend up to 0.7 μm from the surface of the RBC and appear stable enough that we were able to observe them in live cell imaging. To understand how such structures can be generated, we treated RBCs infected with Δ*PFA* parasites with cytochalasin-D (cyto-D) to depolymerise actin, methyl-β-cyclodextrin (MBCD) to chelate membrane cholesterol, or the glucosidases hyaluronidase (HA) and neuraminidase (NA) to degrade RBC glycocalyx. Following treatment, cells were fixed and prepared for SEM, followed by image acquisition and analysis. Neither cyto-D nor HA/NA treatment caused a statistically significant reduction in the number of eKnobs in cells infected with Δ*PFA* (S6A, S6B and S6E Fig). Treatment with MBCD, while not causing a total reversion of eKnobs to knobs, did cause a statistically significant alteration in the observed type of eKnobs, with a decrease in the number of deformed, elongated eKnobs, and an increase in abnormal, enlarged eKnobs (S6C and S6D Fig).

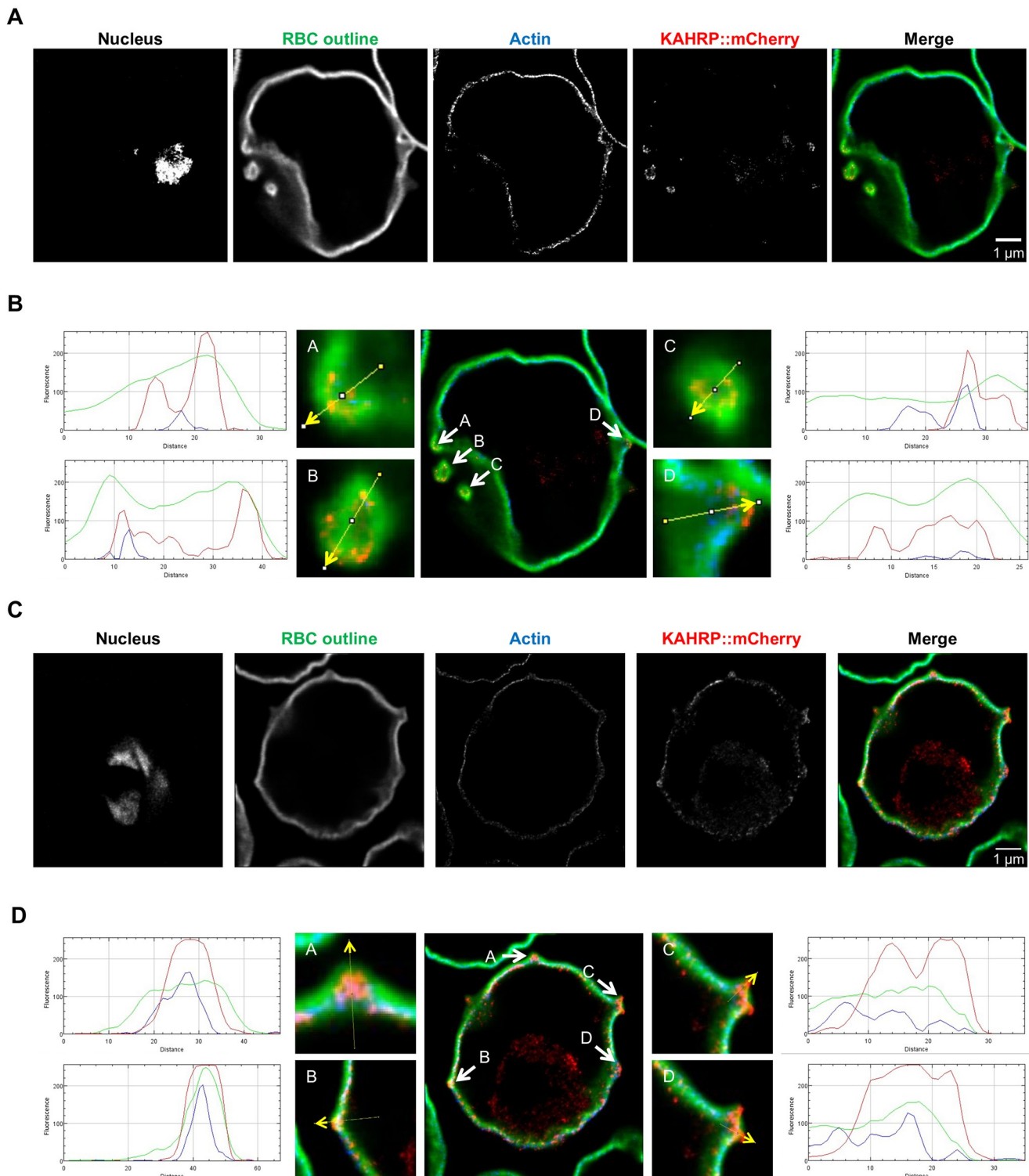

**Fig 4. Investigation of the subcellular composition of deformed knobs in Δ*PFA*[KAHRP::mCherry] via rSTED imaging in an IFA assay.** A, C) DNA was stained using DAPI; WGA was used to stain the RBC glycocalyx; phalloidin was used to stain actin; and RFP booster was used to label KAHRP::mCherry. B, D) Line scan profiles of fluorescence intensity (arbitrary units) along the yellow arrows shown in the fluorescent overlay. The arrows bisect KAHRP_mCherry-rich structures (likely representing knobs). These profiles demonstrate that in the vertical view, phalloidin (*i.e.*, actin) is localised toward the cytosol from the KAHRP structures. The horizontal view shows that the KAHRP-containing structures form a ring structure. These might contain low amounts of actin but are likely filled with other material(s).

## PFA66 truncation results in negligible cytoadhesion and aberrant PfEMP1 presentation

As previously mentioned, mature stage infected *P. falciparum* iRBCs develop additional adhesive capabilities to cells of the host, and this process underlies *P. falciparum* pathogenicity [2]. Cytoadherence of iRBCs to purified ligands can be assessed with a binding assay [21]. Prior to genetic manipulation we selected our parasite population for expression of the *var2CSA* variant of *Pf*EMP1 by repeatedly binding them to CSA [22]. We chose to use the CS2 strain of *P. falciparum* as this strain stably expresses *var2CSA* [21]. A static cytoadherence assay against immobilised CSA demonstrated that, in direct comparison to the parental CS2 strain, Δ*PFA* exhibits massively reduced levels of binding (Fig 5A). Although all experiments were carried out on parasites that had not been maintained in culture for extended time periods (to avoid switching to another var/*Pf*EMP1 variant), we wanted to verify that Δ*PFA* still expressed *Pf*EMP1$^{var2csa}$. We therefore used flow cytometry on intact iRBCs using antisera specific against VAR2CSA (a kind gift of Benoît Gamain) to verify VAR2CSA expression and surface exposure. On RBCs infected with Δ*PFA*, surface expression of VAR2CSA was reduced 60% compared to wild type parasites (Fig 5B and S6F Fig for histograms). IFA using the same antisera on fixed cells demonstrated that both CS2 and Δ*PFA* express VAR2CSA to similar levels, with punctuate staining distributed across the host cell. A control IFA on cells infected with 3D7 strain parasites that had not been selected for VAR2CSA expression showed only background fluorescence, verifying the specificity of the result (S6G Fig).

## Complementation of PFA66 function requires an intact J-domain

HSP40s such as PFA66 are known interactors and regulators of HSP70s [14,15,37], and HSP70-independent functions of HSP40s are rare. The J-domain of HSP40s is crucial both for recruitment of the partner HSP70 and stimulating the ATPase activity of the HSP70 partner [37]. Having shown above that we can achieve functional complementation via episomal expression of a full-length copy of PFA66, we were interested to know whether a functional J-domain (and hence a functional interaction with a HSP70) is required for phenotypic complementation. To this end, we expressed in the Δ*PFA* parasite line a full-length copy of PFA66 with a H111Q amino acid replacement converting the HPD motif of the J-domain into the non-functional QPD sequence (S6H Fig) and assayed knob/eKnob morphology and density in comparison to the wild-type complementation parasites as a surrogate for all other phenotypic assays. Both wild-type (HPD) and mutant (QPD) fusions were expressed at a similar level, and localised to punctuate structures in the iRBC, suggestive of a correct J-dot localisation (S6I Fig). While expression of full-length wild-type PFA66 was able to revert the mutant knob phenotype (Fig 2B and 2C), similar expression of the non-functional QPD mutant failed to complement the WT phenotype (Fig 5C and 5D) in either density or morphology of knobs/eKnobs.

## The J-domain of PFA66 can stimulate both parasite-derived and human Hsp70

Knowing that a functional J-domain is needed to maintain normal knob morphology, we wanted to probe for potential HSP70 partners for PFA66 in the RBC. Residual human HSP70s and one parasite exported HSP70 (*Pf*HSP70-X) are present in the RBC and it has been suggested that the parasite hijacks the RBC chaperone machinery by exporting various JDPs to facilitate remodelling of the RBC [16,18]. To test which of these HSP70s functionally interact with PFA66, we performed single turn-over ATPase assays with *Hs*HSP70 (HSPA1A) and

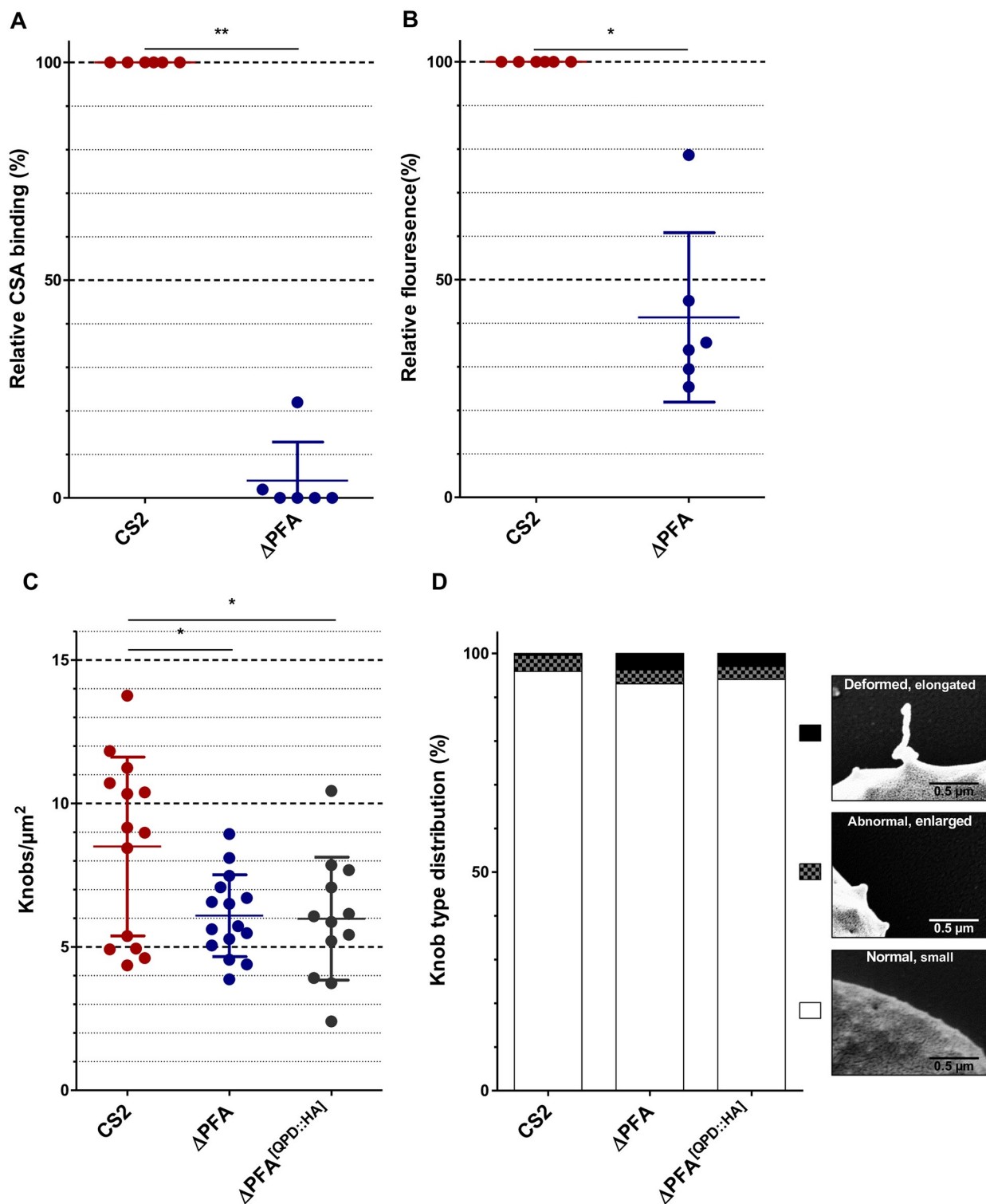

**Fig 5. ΔPFA iRBCs display negligible cytoadherence and lower surface exposed PfEMP1.** A) ΔPFA displays negligible cytoadherence and lower PfEMP1 surface exposure than CS2. CS2 and ΔPFA were assayed to test their ability to adhere to immobilised CSA in Petri dishes using microscopic counting of the cells. Cytoadhesion strength is expressed relative to CS2. Results are shown for six binding assays. B) Analysis of PfEMP1 surface exposure via flow cytometry. IRBCs were stained with DAPI and αVAR2CSA antiserum followed by a Cy3-coupled secondary antibody. ΔPFAs have lower PfEMP1 surface exposure than CS2 in six independent experiments. C, D) Expression of a PFA variant featuring a mutated HPD motif in the cell line ΔPFA[QPD::HA] does not complement reduction in knob abundance (C) and knob deformation (D) observed in ΔPFA. IRBC were purified and imaged via SEM. Knobs were then counted and grouped into three categories.

*Hs*HSC70 (HSPA8), and the parasite encoded *Pf*HSP70-X [38]. Previously, using steady state ATPase assays, it has been shown that PFA66 is able to stimulate the hydrolysis of ATP by *Pf*HSP70-X and (at a lower degree) *Hs*HSC70 but not *Hs*HSP70 [39,40]. The basal ATPase activity of HSP70s is generally very low and stimulated in synergism by J-domain proteins and substrates [37,41]. Certain JDPs are able to provide both signals, the J-domain and the substrate, needed to synergistically stimulate the ATPase activity of HSP70s. To rule out the possibility that the lack of stimulation of *Hs*HSP70 by PFA66 that has been observed by Daniyan et al. [39] is due to PFA66 itself not being a substrate for *Hs*HSP70, we applied the experimental strategy of Landry and colleagues [42] and designed a minimal J-domain-substrate system by fusing a peptide derived from the human heat shock transcription factor 1 (*Hs*HSF1, S461-Q471, referred to as substrate), that has recently been identified as binding site for *Hs*HSC70 [43], C-terminally to the PFA66 J-domain (amino acids 81–148, resulting in PFA66[JDS]). To shield the peptide from degradation, a C-terminal strep-tag (WSHPQFEK) was added to PFA66[JDS] as well. *Hs*HSP70, *Hs*HSC70, *Pf*HSP70-X, and PFA66[JDS] were expressed as HIS-SUMO fusions, purified to homogeneity and used to perform ATPase assays under single turn-over conditions. Under such conditions, the hydrolysis of one molecule of ATP per HSP70 molecule is monitored and other potentially rate limiting processes, such as nucleotide exchange, are neglectable. We observe that PFA66[JDS] is able to stimulate the hydrolysis of ATP by *Hs*HSC70 and *Pf*HSP70-X, as expected, but unexpectedly also by *Hs*Hsp70 (Fig 6A), to significantly higher rates than the peptide alone, supporting the idea that PFA66 is able to recruit and repurpose the two major human HSP70 homologues in RBCs (Fig 6A and 6B).

## Discussion

Although it has long been recognised that malaria parasites export a substantial number of proteins to their host cell, the mature human RBC, the function of many of these proteins remains unknown [7,8,13]. *P. falciparum* exports a larger number of proteins to the host cell than related species, and one family that is highly represented amongst this expansion is that of J-domain proteins (HSP40s [9]). A previous medium-throughput study identified the exported HSP40 PFA66 as likely to be essential and resistant to inactivation via double-crossover integration [12]. In this study, we have used SLI-TGD [20] to generate parasites expressing a severely truncated form of PFA66. This strategy deletes the entire substrate-binding domain of the expressed protein, resulting in a non-functional truncation mutant that is incapable of carrying out its biological function, and is also present at low abundance. The resulting parasites have normal viability but exhibit severe abnormalities in host cell modification.

Parasite growth following truncation of PFA66 was slightly higher than that of wild type CS2 parasites, this effect only becoming evident after four growth cycles. A similar effect was previously observed in a Δ*GEXP07* parasite line; however, the significance of this result remains unclear as the reduction in metabolic burden by the loss of only one gene/protein is likely to be negligible [44]. Lysis of iRBCs in a sorbitol uptake assay was also slightly reduced, implying either a reduction in novel permeability pathway activity, or potentially an increase in host cell stability via mechanical means. As a reduced NPP activity would be expected to lead to slower, not faster, parasite growth, we interpret this to be the result of increased robustness of the RBC plasma membrane through an unknown mechanism. Alternatively, growth in the rich media we used may not have uncovered a more obscure effect on NPP [45], and future studies may wish to address this.

The most striking result of our study was the observation that, in the Δ*PFA66* cell line, normal knob biogenesis was significantly inhibited with regard to both knob density and morphology. Although earlier knockout studies have observed a reduction in knob formation, or

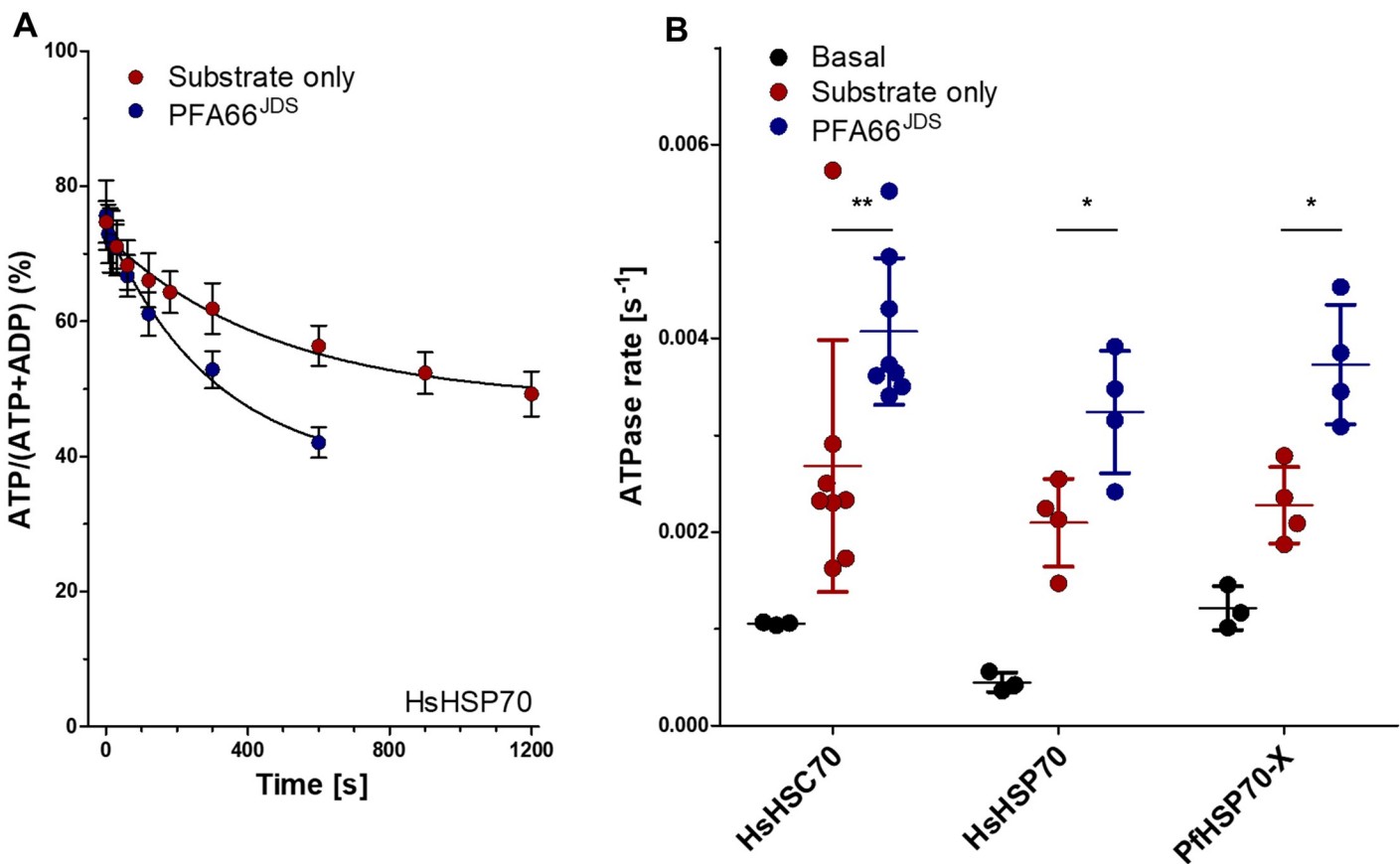

**Fig 6. The J-domain from PFA66 stimulates the ATPase activity of *Hs*HSP70, *Hs*HSC70 and *Pf*HSP70-X.** A) Single turn-over ATP hydrolysis of *Hs*HSP70 in the presence of a peptide substrate (red circles) or PFA$^{JDS}$ (blue circles). Note that in contrast to Daniyan et al. 2016 [39], we clearly observe a stimulation of the hydrolysis of ATP by HsHsp70 (HSPA1A) in the presence of PFA$^{JDS}$ B) Single turn-over ATPase rates for *Hs*HSC70, *Hs*Hsp70 and *Pf*HSP70-X in the absence of any substrates or JDPs (black circles), in the presence of the substrate peptide (red circles) or in the presence of PFA$^{JDH}$ (blue circles). Shown are individual data points and standard deviation of independent experiments (n = 3 to 8). *, p < 0.05; **, p<0.001 (ANOVA with Holm-Sidak's multiple comparison test).

slight alteration in knob morphology, to our knowledge this current study is the first to demonstrate such a dramatic alteration in knob structure upon inactivation of a single gene [12,44]. Indeed, so different are the structures we observe to classic knobs that we suggest calling them eKnob to distinguish them from the 'normal' surface extensions. eKnobs differ from knobs both in their size and length, reaching up to 0.7μm from the RBC surface. Additionally, eKnobs that split into separate branches can be observed. KAHRP, a protein known to be required for correct knob formation, can be localised to eKnobs, although its distribution (based on live cell imaging, IFA, and membrane shearing paired with STED microscopy) seems to be different from that observed in cells infected with wild type parasites. Several studies suggest that KAHRP is integrated into higher order assemblies during parasite development, eventually resulting in a ring structure underlying the knob [46,47]. Deletions in specific KAHRP domains lead to less incorporation into such structures and also appear to influence the generation of sub-knob spiral structures of unknown molecular composition [47]. The possibility exists that KAHRP, while being necessary for correct knob formation, is itself not the major structure-giving component but merely serves as a scaffold for assembly of further higher order molecular structures, which themselves generate the necessary vector force to allow membrane curvature and push the knobs above the surface of the RBC. KAHRP has

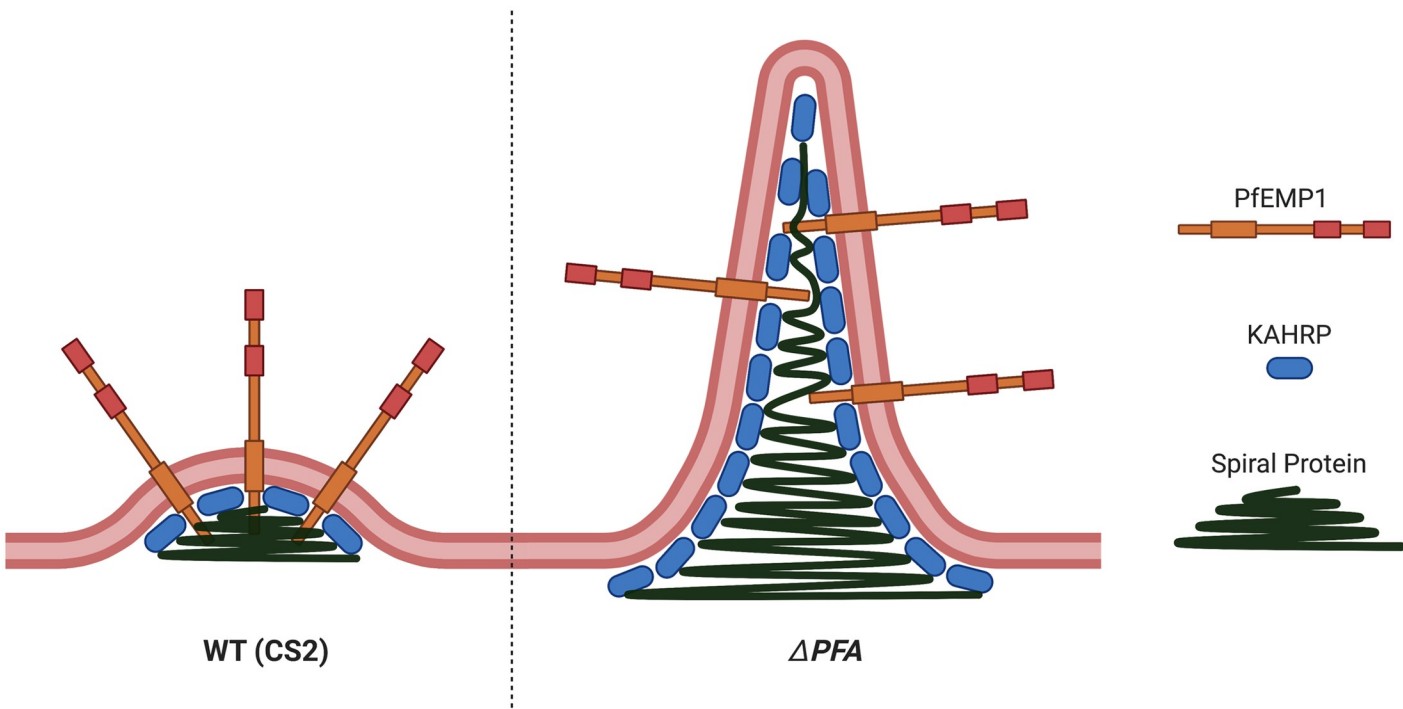

**Fig 7. Proposed model for eKnob formation and structure.** In opposition to normal knob formation in the CS2 cell line (left) runaway extension of the spiral underlying eKnobs in ΔPFA could drive their elongation (right). KAHRP is still present and associated with the inner lumen of eKnobs, PfEMP1 anchored in the eKnobs is incorrectly presented and has thus a reduced cytoadherence capacity.

been proposed to be especially vulnerable to misfolding due to its unusual amino acid composition, which would make it a likely client for chaperones / cochaperone systems [17]. Following this logic, if PFA66 is involved in the assembly of KAHRP into higher order assemblies, interruption of this process could cause knock-on misassembly of the spiral and therefore aberrant knob structures. Although we were not able to visualise spiral structures at the base of eKnobs, our observed phenotype could be explained by runaway lengthening of such a spiral structure (Fig 7). Indeed, the electron density we observed on tomograms is consistent with a high molecular weight structure as a form-giving scaffold for the formation of eKnobs. Within the resolution limit of our study, KAHRP itself appears to line the inner leaflet of the membrane-bounded eKnobs and is thus unlikely to itself be a major component of the electron dense core of the eKnobs. Host actin is known to be required for the generation of knobs [46]; however, although we could successfully visualise actin below the RBC plasma membrane, it appears to be largely excluded from eKnobs and thus cannot be responsible for maintaining the form of these structures, a view supported by the lack of action of cyto-D on eKnob structures. Similarly, enzymatic removal of the RBC glycocalyx had no effect on eKnobs, excluding a role for this in membrane shaping. Chelation of membrane cholesterol via treatment with MBCD did cause change in the morphology of eKnobs, but not a complete reversion to normal knob structures. Removal of cholesterol from biological membranes has been observed to cause an increase in membrane stiffness [48,49], and this may explain the morphological reversion upon MBCD treatment, with a stiffer membrane being more resistant to the pushing force within the eKnobs. Alternatively, removal of cholesterol and subsequent breakdown of so-called lipid rafts may interfere with the higher order organisation of membrane-bound factors involved either directly in membrane curvature or required for coordination of proteins

internal to the eKnobs that generate force. Taken together, our data strongly implies that both knobs and eKnobs contain a so far unknown component, likely proteinaceous, which is required for force generation, subsequent membrane curvature, and resulting morphology. Interruption of correct assembly of this factor leads to aberrations in knob formation, eventually leading to the appearance of eKnobs. Our, and others', data implicates KAHRP as having a role in this process, but it is unlikely to be the only structural protein involved.

In other systems, HSP40s, through their role as regulators of HSP70 chaperone activity, have been shown to have a role in both assembly and disassembly of protein complexes [50], and it is tempting to suggest that the phenotype we observe here is due to incorrect complex assembly. Alternatively, PFA66 may be required for the correct transport of accessory proteins required for complex formation such as those proposed in a recent study [46], and thus play an indirect role in correct assembly of high-molecular weight complexes. In support of this hypothesis, PFA66 is known to associate with J-dots, highly mobile structures within the iRBC that are also known to contain a number of HSP70s [16,18,19,24]. It is also feasible that PFA66 is required for the disassembly of incorrectly folded or assembled knob protein complexes and that our knockout reveals so far unknown quality control mechanisms.

Knobs are required for correct presentation of the major virulence factor *Pf*EMP1, and high affinity binding of such to endothelial receptors [51]. Although previous studies suggested that KAHRP and *Pf*EMP1 formed a 'pre-cytoadherence complex' at the Maurer's clefts [52], later evidence suggest strongly that *Pf*EMP1 is incorporated only into knobs once they have been at least partly formed [46,53]. RBCs infected with Δ*PFA66* parasites showed a 63% reduction in knob/eKnob density and a 10-fold increase in the frequency of abnormal knob phenotype, and we observed an almost total lack of cytoadherence in iRBC infected with Δ*PFA66* parasites. This data strongly supports the view that, even in the normal knobs present, less *Pf*EMP1 was correctly presented and could take part in cytoadherence. Flow cytometry determined an almost 60% drop in cell surface recognition of the VAR2CSA variant of *Pf*EMP1, although IFA suggests that both cell lines express similar amounts of this protein. The total loss of cytoadherence may be the consequence of several distinct factors: a) less total surface *Pf*EMP1, b) fewer knobs with correctly loaded *Pf*EMP1, and c) a significant number of aberrant knobs/eKnobs. Moreover, we cannot exclude that *Pf*EMP1, which needs to be correctly folded to bind specific receptors, does not assume the correct tertiary structure due to the lack of the necessary chaperone-cochaperone system.

As previously mentioned, HSP40s generally act in concert with members of the HSP70 family [15]. The RBC is known to contain significant amounts of residual human HSP70s [54] and additionally a parasite-encoded HSP70, *Pf*HSP70-X [18]. We have previously demonstrated that a knockout of *Pf*HSP70-X leads to a reduction in virulence characteristics, including cytoadherence [55]. Significantly, however, iRBCs infected with Δ*70-X* parasites were covered with normal knob structures at a density comparable to that of the wild type [55] and a further study even suggests that deletion or down-regulation of *Pf*HSP70-X leads to no observable mutant phenotype [56]. Hence the mutant phenotype of RBCs infected with Δ*PFA66* is distinct and significantly more dramatic than that in iRBCs infected with Δ*70-X* parasites. Considering the difference in phenotype between Δ*70-X* and Δ*PFA66*, here we must conclude that, if a HSP70 is involved, it is more likely to be of human rather than parasite origin, a possibility also previously raised by others [56].

The J-domain of HSP40s contains a characteristic HPD motif that is required both for binding and ATPase activation of partner HSP70s [50]. As the unlikely possibility existed that our observed phenotype was due to a HSP70-independent function of PFA66, we carried out complementation analysis with a full-length copy of PFA66 expressed from an episome, or a copy containing a H111Q mutation that renders the J-domain inactive. Despite similar expression

levels and correct localisation of both fusions, only the wild type protein was able to complement the mutant knob phenotype, clearly demonstrating that recruitment of Hsp70 by PFA66 is essential for the wild type phenotype. We therefore conclude that the effects we observed upon PFA66 truncation are due to either a negative effect of the truncated protein on proper functioning of the *Hs*HSP70 chaperone system within the host RBC or result from a lack of cochaperone activity via deletion of an essential functional domain (SBD) of the HSP40. Either way the results furthermore support *Hs*HSP70 involvement.

To regulate the function of residual human HSP70s, the J-domain of PFA66 should be able to stimulate the ATPase activity of these proteins, however several studies have demonstrated only weak functional interactions. One potential weakness of these published studies is the failure to distinguish between chaperone-cochaperone and chaperone-substrate interactions, which may produce misleading results. To throw more light on this topic, we uncoupled these interactions and reconstituted a minimal J-domain system by fusion of a suitable substrate peptide to the J-domain derived from PFA66. Single-turnover ATPase assays revealed significant stimulation of both the parasite derived *Pf*HSP70-X and human HSP70s compared to controls, further supporting our assertion that the mutant phenotype observed upon deletion of PFA66 is linked to the activity of residual human HSP70. We cannot formally exclude that PFA66 functionally interacts with both human and parasite HSP70 homologues and that the phenotype we observe is a combination of the effect on both chaperones, but the balance of probabilities suggests that morphological abnormalities observed in this current study are largely due to an interruption of functional PFA66-*Hs*HSP70/HSC70 interactions.

A potential role for residual human HSP70 in host cell modification and parasite virulence has been suggested for almost 20 years [57], but to our knowledge our current study is the first to provide strong experimental evidence directly implicating human HSP70s in these processes. We wish to note that we cannot unequivocally state that our complementation construct caused reversion of all the aberrant phenotypes herein investigated, however we feel it likely that all abnormal phenotypes are linked and thus our "proof of principle" investigation of knob/eKnob phenotypes is likely to be reflected in other facets of host cell modification and can thus be seen as a proxy for such.

To conclude, in this study we show data suggesting that correct biogenesis of knobs in malaria-iRBCs is a complex process necessitating a number of proteins, the molecular identity of some of which remains enigmatic. Our data suggests that KAHRP, while obviously required for knob generation, may not directly provide a scaffold for knob structure. More importantly, our data also reveals that residual human HSP70 within the iRBC is involved in parasite-driven host cell modification processes. To our knowledge, this is the first time a host cell protein has been directly implicated in parasite virulence and protein transport processes, and this observation opens up exciting new avenues not only for intervention strategies, but also for the discovery of other novel cell biological phenomena.

## Materials & methods

### Vector construction

The ~1kbp PFA66 targeting region was amplified using the primers PFA_NotI_F and PFA_MluI_R and cloned into pSLI$^{TGD}$ (kind gift of Tobias Spielmann [20]) using NotI-HF and MluI (NEB). The complementation plasmid PFA::HA was generated by excising the *PFA0660w* coding sequence and promoter from the plasmid pAD-A660-GFP [16] with the restriction enzymes NotI-HF and BssHII and cloning them into pARL-PFF1415c-3xHA (BSD, a kind gift of Sarah Charnaud). The resulting plasmid was subsequently used to generate the QPD::HA plasmid via QuickChange PCR using PFA_Quick_QPD_F and PFA_Quick_QPD_R primers.

Upon verification of the QPD mutation, the insert consisting of the *PFA0660w* promoter and coding sequence was re-cloned into the same vector to avoid mutation due to the PCR step. The KAHRP::mCherry plasmid was generated by amplifying mCherry with the primers mCherry_AvrII_F and mCherry_XmaI_R and cloning them into a pre-existing plasmid pARL2_KAHRP containing the native KAHRP promoter and KAHRP coding sequence (kind gift of Cecilia Sanchez). All primers are listed in S2 Table.

## Cell culture methods

*P. falciparum* parasites were cultured at 37˚C with 90% N2, 5% $CO_2$, and 5% $O_2$ according to established methods [58]. Parasites were maintained at a haematocrit of ~5% in $A^+$ or $O^+$ human blood obtained from the blood banks in Marburg and Heidelberg and maintained with RPMI1640 (Gibco) containing 200 µM hypoxanthine, 160 µM neomycin (Sigma Aldrich) and 10% human plasma. Parasitaemias were evaluated from smears prepared from the blood cultures, which were fixed in 100% MeOH and stained with $ddH_2O$/10% Giemsa solution (Merck). Parasites were transfected with 150 µg of plasmid and treated with 2.5nM WR99210 (HS lines) or 12µg/ml blasticidin (Invivo Gen). Transfectants were propagated in fresh $O^-$ blood using RPMI1640 (Gibco) with 5% human plasma, 5% Albumax II (Invitrogen) and other additives as above until parasites re-appeared [59]. Selection-linked integration was performed according to Birnbaum et al. [20]. Briefly, following reappearance after the initial transfection, parasites were treated with 400 µg/ml G418 (Thermo Fisher Scientific) until resistant parasites were observed. Parasites were synchronized before experiments using sorbitol-induced lysis [60]. For this, mixed-culture parasites were incubated in 5% sorbitol for 10 min, washed with parasite culture medium and re-cultivated. Routine selection for CSA-binding parasites was performed according to standard protocols [22]. Late-stage parasites were enriched via gelatine flotation for 1 hr / 37˚C [61]. Subsequently parasites were resuspended in cytoadhesion media (pH 7.2, prepared from RPMI1640 powder, Life technologies) and incubated in cell culture flasks pre-treated with CSA (PBS pH 7.2/1 mg/ml CSA overnight / 16˚C and blocked with PBS pH 7.2 / 1% BSA) for 1 hr / 37˚C. After careful washing with cytoadhesion media, the remaining bound parasites were washed off and re-seeded.

## Chemical or enzymatic treatment of iRBCs

For these experiments, magnetically purified iRBCs (~1 x $10^7$ per condition) were used. Cytochalasin-D treatment was carried out with RPMI1640 / 10 µM cyto-D for 10 min at RT, incubation with RPMI1640/10 mM MBCD was performed for 20 min at 37˚C, RPMI1640/ 30mU neuraminidase or RPMI1640/30U hyaluronidase treatments were performed for 1 hr at 37˚C. Following treatment, samples were processed for SEM.

## MACS purification

For some protocols, late-stage parasites were magnetically purified using a VARIOMACS with a CS-column. Briefly ~1 ml packed RBCs (~10% parasitaemia) were applied to a CS column, washed with PBS / 3% BSA and finally eluted into PBS.

## Microscopy methods

Live cell imaging was performed on DAPI-stained (1 ng/ml) parasites using a Zeiss Axio-Observer microscope and AxioVision software. For IFA assays, parasites were fixed on microscopy slides using 90% acetone / 10% MeOH for 5 min / -20˚C. Cells were then blocked using PBS / 3% BSA for 1 hr / RT and incubated in a humid chamber overnight with the primary

antibody diluted in blocking buffer (for antibodies see S3 Table). On the next day, PBS-washed slides were treated with the secondary antibody, diluted 1:2000 in blocking buffer for 2 hr / RT, subsequently washed again, DAPI stained (0.1 ng/ml in PBS), and imaged using a Zeiss Axio-Observer microscope and AxioVision software. RSTED imaging was carried out as recently reported in great detail [62]. Wheat germ agglutinin Alexa Fluor 488 conjugate (Thermo Fisher) was used according to the supplier's specification. Phalloidin–Atto 647N (Thermo Fisher) was diluted 1:500 in PBS and incubated for 30 min / RT to stain the actin cytoskeleton. RFP booster ATTO594 was used 1:200/2hr/RT to enhance RFP fluorescence. IMSpector imaging software (Abberior Instruments GmbH) was used for image capture and deconvolution of STED images, and AxioVision software was used for other acquisitions. Images were processed using ImageJ. Brightness and contrast were adjusted to reduce background and enhance visibility. No gamma adjustments were applied to any images, and all data is presented in accordance with the recommendations of Rossner and Yamada [63].

## Protein-based methods

Protein extracts were prepared from 1 x $10^8$ MACS-purified iRBCs. These were resuspended in PBS and boiled in Laemmli loading buffer for 10 min at 99°C. Soluble fractions were separated via centrifugation (4°C, 35,000 g) and an equivalent of 1 x $10^7$ parasites loaded onto each well of 12% acrylamide gels. Equinatoxin (EQT) treatment and fractionation of MACS-purified iRBCs was carried out as described by Külzer et al. [27] but using 4 haemolytic units of EQT at RT for 6 min. Western blot / immunodetection was carried out via semi-dry blotting, blocking in 5% milk powder (1 hr / RT), incubation with primary (overnight / 4°C), washing three times with PBS, incubation with the secondary (2 hr / RT) antibody in blocking buffer, washing three times with PBS, and visualization via x-ray films. Antibody sources and dilutions can be found in S3 Table.

## Membrane shearing

For investigation of the internal structure of the RBC cytoskeleton membrane shearing was employed according to established protocols [46,64]. Briefly, a (3-aminopropyl)triethoxysilane-treated ibidi dish was incubated with 150 μl PBS / 1 mM with Bis(sulfosuccimidyl)suberate for 30 min / RT, washed with PBS, and incubated with 150 μl ddH$_2$O / 0.1 mg/ml erythroagglutinating phytohaemagglutinin for 2 hr / RT. Dishes were rinsed three times with PBS and quenched using PBS / 0.1 M glycine for 15 min / RT. Approximately 1 x $10^7$ MACS-purified iRBCs were added and incubated for 3–4 hr, washed, and sheared using 5P8-10 buffer (5 mM Na$_2$HPO$_4$ / NaH$_2$PO$_4$, 10 mM NaCl, PH 8), while angling the dish at 20°. Samples were then blocked using 150 μl of PBS / 1% BSA, treated with the primary α-KAHRP antibody overnight / 4°C in blocking buffer, washed three times, incubated with the secondary α-mouse[ATTO549] for 1 hr / RT in blocking buffer, and finally washed three times before imaging via rSTED.

## Sorbitol lysis

Assessment of NPP activity was carried out according to Baumeister et al. [28]. For each measurement, 40 μl of 2% trophozoite culture was resuspended in 150 μl lysis buffer (290 mM sorbitol, 5 mM HEPES, pH 7.4) and incubated for 30 min / 37°C. Remaining RBCs were then pelleted at 1,600 g / 2 min, and the absorbance of the resulting supernatant was measured at OD$_{570nm}$. The highest value obtained in a biological replicate was set to 100% to ease comparison of datasets We carried out 10 biological and 4 technical replicates.

## Flow cytometry

IRBCs were fixed for 24 hr at 4˚C using PBS/4% paraformaldehyde/0.0075% glutaraldehyde and stained with DAPI (1 ng/ml) prior to analysis with a BD Canto. In the growth experiments, both cell lines were diluted after every growth cycle with the same factor in order to support parasite growth. Both parasite cell lines were seeded with the same parasitaemia and diluted after every cycle to avoid 'crashing' the culture. Parasitaemias were measured before and after every dilution by staining of iRBCs with DAPI and flow cytometry. For staining VAR2CSA on the RBC surface, live parasites were incubated with VAR2CSA antiserum (11P, rabbit, a kind gift of Benoit Gamain) and α-rabbit-Cy3 for 30 min each and then processed for flow analysis as detailed above.

## Cytoadherence

IRBC cytoadhesion to immobilised CSA was investigated using MACS-purified late-stage parasites [21]. Parasites were applied in cytoadhesion media (pH 7.2, made from RPMI1640 powder, Life technologies) to pre-treated spots (PBS pH 7.2/1 mg/ml CSA overnight at 16˚C, blocked with PBS pH 7.2/1% BSA for 1 hr at RT), and then washed with PBS on a Petri dish. After incubation for 1 hr at RT, non-bound parasites were washed away using cytoadhesion medium. Parasites were then fixed using PBS/2% glutaraldehyde for 2 hr at RT and stained with PBS/10% Giemsa for 10 min at RT prior to imaging using a Zeiss Axio Observer microscope and counting with Ilastik [33] and ImageJ software.

## Electron microscopy

For scanning electron microscopy, purified parasites were fixed using PBS /1% glutaraldehyde for at least 30 min at RT. After washing, parasites were bound to coverslips (pre-treated with 0.1% polylysine for 15 min at RT), washed again, and dehydrated in acetone gradients (ddH$_2$O, 25% Ac, 50% Ac, 75% Ac, 100% Ac, 10 min each) followed by critical point dehydration and coating with 5 nm Pd-gold. Cells were imaged using a Zeiss Leo 1530 electron microscope (SE2 detector, ~12,000 x magnification [65]. For transmission electron microscopy, parasites were fixed in 100 mM Ca-cacodylate/4% paraformaldehyde/ 2% glutaraldehyde, embedded in Spurr and cut into ~70 nm sections. Some samples were fixed using 100 mM Ca-cacodylate/4% paraformaldehyde/ 0.1% glutaraldehyde and treated according to the Tokuyasu protocol for immunogold labelling of KAHRP using an α-KAHRP (rabbit) antibody and a secondary goat α-rabbit-gold conjugated antibody [66]. Some of the EM sections were used without post-contrasting, while some were post-contrasted using 3% uranyl acetate and ddH$_2$O/ 0.15 M Na-citrate/0.08 M Pb(NO$_3$)$_2$ / 0.16 M NaOH for 2 min. Imaging was performed using a Jeol 1400 microscope operating at 80kV. For electron tomography ~350 nm thick sections were used and examined in a TECNAI F30, 300kV FEG, FEI electron microscope (EMBL Heidelberg). The resulting tomograms were processed using IMOD, ETOMO image/ volume processing software package and the Amira, volume visualisation software.

## Statistics

Statistics were calculated in prism or Excel using unpaired, two-tailed t-tests. $p > 0.05$ = non-significant (ns); *$p < 0.05$; **$p < 0.01$; ***$p < 0.00$. Figs show mean and standard deviation.

## ImageJ macro

The ImageJ/Fiji [67] macro computes the local maxima of each object on the smooth probability map (PM) images generated by ilastik pixel classification. The ilastik [33] pixel classification

workflow is used to reduce the background in the images and enhance the foreground pixels. To segment each object in the probability map images the local maxima is used as a seed for the 3D watershed plugin. The approach allows to separate close objects and creates masks that are used to measure size and intensity on the raw images. The macro and the instructions on how to use it can be found at: https://github.com/cberri/2D_AutomatedObjectsDetection_ImageJ-Fiji

### Protein expression and purification

The J-domain fusion construct, PFA$^{JDS}$ used in this work was expressed in the *E. coli* BL21 Rosetta strain as N-terminal SUMO fusion protein with a C-terminal Strep-tag (WSHPQFEK) and incubated at 30˚C during all steps of expression. Multiple transformed colonies were used to inoculate 100 ml of 2xYT medium supplemented with kanamycin (50μg/ml) and 25 ml of overnight culture used the next day to inoculate 3 l of 2xYT medium containing kanamycin. The culture was grown until an $OD_{600}$ of 1.2 was reached and induced with 1 mM IPTG and the protein was expressed for 2h. Cells were then harvested at 5000 g for 15 min at RT and cell pellets resuspended in 25 ml of chilled lysis buffer (50 mM Tris/HCl pH 7.9, 500 mM NaCl, 0,6% Brij58, 1 mM PMSF, 5 mM $MgCl_2$, 10 μg/ml DNaseI, 8 μg/ml pepstatin, 10 μg/ml aprotinin and 5 μg/ml leupeptin) per 1.5 l of culture and snap frozen in liquid nitrogen and stored at -20˚C. All subsequent steps were carried out at 4˚C or on ice if not indicated otherwise. Cell lysis was performed by passing the cells two times through a MicroFluidizer (Avestin, Ottawa, Canada) after fresh proteinase inhibitors were added. The obtained lysate was clarified by centrifugation at 20.000 g for 30 min. The supernatant was mixed with 2 g of protino beads (Ni-IDA, Macherey-Nagel, Düren, Germany) and incubated for 20 min rotating. The protino was subsequently collected in a gravity-flow column and washed first with 12.5 column volumes (CV) of wash buffer (50 mM Tris/HCl pH 7.9, 500 mM NaCl, 0.1% Brij58 and 2 M Urea) and then with 12.5 CV of wash buffer with 1.5 M NaCl. The protein was then washed with 12.5 CV elution buffer (50 mM Tris/HCl pH 7.9, 500 mM NaCl, 2 mM 2-mercaptoethanol and 2 M Urea) to remove detergent and afterwards supplemented with another 6 ml of elution buffer, the flow stopped and 8 mg of Ulp1 (produced in the Mayer laboratory) were added and the column was incubated rotating for 2 h. After 2hs of incubation, 0.32 g of fresh protino was added and incubated for 20 mins to remove Ulp1. After Ulp1 removal, the supernatant was collected and the protein dialyzed against 2 l dialysis buffer (40 mM HEPES/KOH pH 7.6, 300 mM KCl, 2 β-Mercaptoethanol and 10% glycerol) overnight. The next day, the protein was aliquoted and snap frozen in liquid nitrogen and stored at -80˚C. Human HSC70(HSPA8)/HSP70(HSPA1) were purified as His6-SUMO fusions after overproduction in *E. coli* BL21 (DE3) Rosetta. Cells were resuspended in Hsp70 lysis buffer (50 mM Tris/HCl pH 7.5, 300 mM NaCl, 5 mM $MgCl_2$, Saccharose 10%) supplemented with 3 mM β-mercaptoethanol, DNase I and protease inhibitors (10 μg/ml aprotinin, 10 μg/ml leupeptin, 8 μg/ml pepstatin, 1 mM PMSF) and lysed using a chilled microfluidizer (Avestin, Ottawa, Canada). The resulting lysate was centrifuged (33000 x g, 4˚C for 30 min) and the supernatant incubated for 25 min at 4˚C with 1.5 g Protino Ni-IDA resin (Macherey Nagel) on a rotation shaker. Afterwards, the resin was transferred to a gravity-flow column, washed with 200 ml of Hsp70 lysis buffer, followed by an ATP wash with 50 ml ATP buffer for 30 mins (Hsp70 lysis buffer with 5 mM ATP). Subsequently, the ATP was removed by washing with 50 ml lysis buffer and the bound protein was eluted with Hsp70 lysis buffer containing 300 mM imidazole, 3 mM β-mercaptoethanol and protease inhibitors. Protein containing fractions were pooled, dialyzed against HKMG150 buffer (25 mM HEPES/KOH pH 7.6, 150 mM KCl, 5 mM $MgCl_2$, 10% glycerol) and digested with 400 mg of Ulp1 overnight. After proteolytic cleavage, the protein was

incubated with 1.5 g Protino Ni-IDA resin (Macherey Nagel) for 25 min at 4˚C to remove His6-SUMO. Flowthrough containing Hsc70/Hsp70 was buffer exchanged into HKMG10 (25 mM HEPES/KOH pH 7.6, 10 mM KCl, 5 mM MgCl$_2$, 10% glycerol) using a HiPrep 26/10 column and further purified using an 1 ml RESOURCE Q column (10–1000 mM KCl in 16 CV in HKMG buffer) on an ÄKTA purifier system (Cytiva). Fractions containing the target protein were collected, concentrated to around 50 μM, aliquoted, flash-frozen in liquid nitrogen and stored at 80˚C. PfHSP70-X was expressed and purified as previously described [39].

### ATPase assays

The ATPase activities of *Hs*HSC70, *Hs*HSP70 and PfHSP70-X were determined under single-turnover conditions as previously described [38]. The HSP70–ATP complexes (final volume 52 μl) were formed by mixing 50 μl of 30 μM HSP70 with 2 μl of ATP-mix (20mM ATP, 12 μCi [α$^{32}$P]-ATP) in reaction buffer (25 mM HEPES/KOH pH 7.6, 50 mM KCl, 10 mM MgCl$_2$, 2 mM DTT) and leaving the mixture for 2 min on ice. The complexes were separated from unbound ATP at RT by gel filtration on NICK columns (Cytiva), pre-saturated with 1 ml of a BSA solution (1 mg/ml) and pre-equilibrated with 3 column volumes of chilled reaction buffer. Twenty fractions (one drop per fraction) were collected on ice in cooled tubes. The first four fractions containing radioactivity as detected by Geiger counter were pooled, divided into 6.5 μl aliquots, snap frozen in liquid nitrogen and kept at -80˚C. For the ATPase activity determination, HSP70–ATP complex solutions were thawed by hand and, after withdrawal of 0.5 μl for the zero-time point, mixed with 44 μl reaction buffer containing either no cofactors or 1 μM pepHsf1(S461-Q471) or 1 μM PFA$^{JDS}$ protein. At given time points, 2 μl were withdrawn from the reaction mixture and spotted onto a thin layer chromatography plate (PEI Cellulose, Merck, Darmstadt, Germany), which was pre-spotted with 5 μl of a mixture of 5 mM ADP and 5 mM ATP. Developing the plates in 400mM LiCl in 10% acetic acid separated ADP from ATP. Dried plates were exposed to fluoroimaging screens overnight and relative amounts of ATP and ADP were quantified by Fuji FLA 2000 Phosphoimager.

## Supporting information

**S1 Fig. dPFA::GFP fusion protein is exported to the iRBC.** A) Upper panel, schematic of integration PCR strategy; Lower, integration PCR in original uncropped version. Table shows primer combinations and predicted result. B) An equinatoxin lysis experiment demonstrates export of truncated PFA::GFP. Equinatoxin (EQT) treatment selectively lyses the RBC membrane but leaves the PVM and PPM intact. Consequentially, parasite proteins exported to the host cell are found in the supernatant, while other parasite proteins remain in the pellet. Detection of the PV protein SERP and the parasite protein ALDO in the pellet fraction demonstrates intactness of the PV membrane and PPM, respectively. Truncated PFA::GFP was detected alongside human HSP70 in the supernatant fraction, demonstrating its export to the iRBC. C) Δ*PFA* display a slight decrease in NPP activity when compared to CS2. iRBCs were incubated with the hypotonic agent sorbitol, and NPP activity was assessed by measuring the OD of the supernatant. Results are shown for ten biological replicates. D) Growth of CS2 and Δ*PFA* was measured over four cycles via flow cytometry of DAPI-stained, fixed parasites. Δ*PFA* show a slight growth advantage over CS2 in the last cycle. Results are shown for three independent experiments. (TIF)

**S2 Fig. SEM gallery.** SEM image of a non-iRBC and additional SEM images of CS2, Δ*PFA* and Δ*PFA*$^{[PFA::HA]}$. Included is a display of aberrant eKnob morphologies. (TIF)

**S3 Fig. Additional TEM images of CS2 and ΔPFA.** Marked areas are enlarged in the bottom panels to show Maurer´s cleft morphology in more detail.
(TIF)

**S4 Fig. IFA localisation of exported proteins.** Investigation of marker protein localisation using specific antisera in MeOH acetone-fixed ΔPFA with an IFA assay. No drastic difference in the localisation of EXP2, HSP70x, SBP1, REX2, PFEMP3, or PHISTC was found.
(TIF)

**S5 Fig. KAHRP distribution within eKnobs.** A) Verification of CS2$^{[KAHRP::mCherry]}$ and ΔPFA$^{[KAHRP::mCherry]}$ via Western blot verifies the production of KAHRP::mCherry protein in the cell lines using an α-mCherry antibody (expected MW 92kDa). The parasite protein aldolase (ALDO) was used as a loading control. B) Investigation of label distribution in the α-KAHRP immuno-TEM. Distance of label from the base of the knob was measured using ImageJ and expressed relative to the length of the entire knob in percentages (0% being the base and 100% the top). The distribution of label along the full length of the knobs did not differ between the strains and knob types, C) Distance of label to the closest membrane was measured using ImageJ, revealing no difference between the strains and knob types. D) RSTED images of CS2$^{[KAHRP::mCherry]}$ reveal close association of KAHRP::mCherry with the cytoskeleton and glycocalyx. Larger aggregates of KAHRP::mCherry were, in contrast to ΔPFA$^{[KAHRP::mCherry]}$, not observed.
(TIF)

**S6 Fig. Investigation of chemical or enzymatic treatment of iRBCs on eKnobs.** A) Treatment with the actin depolymerizing agent cytochalasin-D does not resolve knob density (A) and eKnob morphology (B, N = 15). Investigation of treatment with the lipid raft disruptor MBCD (C, D) the glucosidases hyaluronidase (HA) and neuraminidase (NA) (E) on knob-type distribution in the two cell lines. N = 15. F) Concatenation of all iRBC data from the experiments in Fig 5B also shows a decrease in surface exposed in ΔPFA across all experiments. Total number of single cells: 1,798,268 (CS2) and 1,799,274 (ΔPFA). G) Investigation whether MeOH-fixed parasites with an α-VAR2CSA antibody demonstrate that both CS2 and ΔPFA express var2CSA to similar levels. H) Comparison of episomally expressed trans-gene (PFA:: HA, QPD::HA) expression levels in ΔPFA. The parasite protein SERP was used as a loading control. I) Immunofluorescence verifies export of episomally expressed PFA::HA and QPD:: HA fusions to the host cell.
(TIF)

**S7 Fig. Full-length versions of all Western blots.** A) Western blot verifies truncation of *PFA0660w* in ΔPFA [Fig 1]. B) Verification of the complementation cell line ΔPFA$^{[PFA::HA]}$ [Fig 1D]. C) Equinatoxin experiment demonstrates export of the dPFA::GFP fusion protein to the iRBC [S1A Fig]. D) Verification of KAHRP::mCherry expression in CS2$^{[KAHRP::mCherry]}$ and ΔPFA$^{[KAHRP::mCherry]}$ [S5C Fig]. E) Comparison of episomally expressed trans-gene (PFA:: HA, QPD::HA) expression levels in ΔPFA [S6H Fig].
(TIF)

**S1 Video. 3D reconstruction and surface render of eKnob depicted in Fig 2E and 2F.** Red, RBC plasma membrane; blue, electron dense material.
(MOV)

**S2 Video. Z-stack of RBCs infected with CS2$^{[KAHRP::mCherry]}$ in mCherry channel.**
(AVI)

**S3 Video. Z-stack of RBCs infected with CS2[KAHRP::mCherry] in mCherry channel.**
(AVI)

**S4 Video. Z-stack of RBCs infected with Δ*PFA*[KAHRP::mCherry] in mCherry channel.**
(AVI)

**S5 Video. Z-stack of RBCs infected with Δ*PFA*[KAHRP::mCherry] in mCherry channel.**
(AVI)

**S1 Table. Predicted exported *P. falciparum* HSP40s and previously observed phenotypes.**
(XLSX)

**S2 Table. Sequences of primers used in this study.**
(XLSX)

**S3 Table. Antibody sources and usage conditions.**
(XLSX)

## Acknowledgments

We wish to thank the blood banks of the University Hospitals in Giessen and Marburg for providing blood. Further we would like to thank the EMCF at the University of Heidelberg (in particular Stefan Hillmer), the EMCF of EMBL Heidelberg (Yannick Schwab and Martin Schorb), the FACS core facility at the ZMBH (Monika Langlotz), Moritz Koch for purifying *Hs*HSP70 and *Hs*HSP70, as well as the Infectious Diseases Imaging Platform (Vibor Laketa) and Marie Freudenberg. The great generosity of colleagues who provided reagents (noted in the text) should be recognised. We thank Tim Bostick for proofreading.

## Author Contributions

**Conceptualization:** Cecilia P. Sanchez, Matthias P. Mayer, Jude M. Przyborski.

**Formal analysis:** Jude M. Przyborski.

**Funding acquisition:** Jude M. Przyborski.

**Investigation:** Mathias Diehl, Lena Roling, Lukas Rohland, Sebastian Weber, Marek Cyrklaff, Cecilia P. Sanchez, Carlo A. Beretta, Caroline S. Simon, Julien Guizetti, Julia Hahn, Norma Schulz, Matthias P. Mayer, Jude M. Przyborski.

**Methodology:** Marek Cyrklaff, Carlo A. Beretta, Caroline S. Simon, Julien Guizetti, Jude M. Przyborski.

**Project administration:** Jude M. Przyborski.

**Supervision:** Matthias P. Mayer, Jude M. Przyborski.

**Writing – original draft:** Mathias Diehl, Jude M. Przyborski.

**Writing – review & editing:** Mathias Diehl, Lukas Rohland, Matthias P. Mayer, Jude M. Przyborski.

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
