## [Editor Report · Decision Letter 0]

28 May 2021

Dear Dr Przyborski,

We received your recent submission to Reviews Commons, along with the reviewers' critiques, for consideration for publication in PLOS Pathogens. In light of the generally positive reviews and the appropriateness of the subject, we would like to invite the resubmission of a significantly-revised version that takes into account the reviewers' comments.

We cannot make any decision about publication until we have seen the revised manuscript and your response to the reviewers' comments. Your revised manuscript is also likely to be sent to reviewers for further evaluation.

Sincerely,

Kirk W. Deitsch

Section Editor

PLOS Pathogens

Kirk Deitsch

Section Editor

PLOS Pathogens

Kasturi Haldar

Editor-in-Chief

PLOS Pathogens

orcid.org/0000-0001-5065-158X

Michael Malim

Editor-in-Chief

PLOS Pathogens

orcid.org/0000-0002-7699-2064
---

## [Decision Letter · Decision Letter 1]

13 Sep 2021

Dear Dr Przyborski,

Thank you very much for submitting your manuscript "Co-chaperone involvement in knob biogenesis implicates host-derived chaperones in malaria virulence" for consideration at PLOS Pathogens. As with all papers reviewed by the journal, your manuscript was reviewed by members of the editorial board and by several independent reviewers. The reviewers appreciated the attention to an important topic. Based on the reviews, we are likely to accept this manuscript for publication, providing that you modify the manuscript according to the review recommendations.

As you will see from the written critiques, all three reviewers were largely satisfied with the revised version of the manuscript that you submitted. However, please address the minor suggestions and comments that the reviewers provided (no additional experiments are requested). 

Sincerely,

Kirk W. Deitsch

Section Editor

PLOS Pathogens

Kirk Deitsch

Section Editor

PLOS Pathogens

Kasturi Haldar

Editor-in-Chief

PLOS Pathogens

orcid.org/0000-0001-5065-158X

Michael Malim

Editor-in-Chief

PLOS Pathogens

orcid.org/0000-0002-7699-2064

Reviewer Comments (if any, and for reference):

Reviewer's Responses to Questions

**Part I - Summary**

Reviewer #1: In this paper the function of Plasmodium falciparum exported protein PFA66, is investigated by replacing its functionally important dnaJ region with GFP. These modified parasites grew fine but produced elongated knob-like structures, called eknobs, at the surface of the parasites infected RBCs. Knobs are elevated platforms formed by exported parasite proteins at the surface of the infected RBC that are used to display PfEMP1 cytoadherance proteins which help the parasites avoid host immunity. The eknobs still display some PfEMP1 and contain exported proteins such as KAHRP but can no longer facilitate cytoadherence. Complementation of the truncated PFA66 with full length protein restored normal knob morphology however complementation with a non-functional HPD to QPD mutant did not restore normal morphology implying interaction of the PFA66 with a HSP70 possibly of host origin is important for function. The authors investigate the potential interaction of PFA66 with two human RBC HSP70s and an exported HSP70x of parasite origin by showing that a recombinant PFA66 fused to a HSP70 substrate peptide can stimulate the ATPase activity of the 3 HSP70s. It has long been suspected that because parasites export many dnaJ domain proteins into their RBCs that they must be exploiting human HSP70s to promote parasite virulence and survival. This paper now not only shows this is likely but also alludes as to what specific functions ie, knob formation, that the PFA66 protein is involved in.

Reviewer #2: This manuscript by Diehl et al reports on the function of the exported J-domain protein PFA66 in remodeling the infected RBC. The authors have largely resolved my earlier concerns and have also provided new data that indicate PFA66 is able to stimulate the ATPase activity of both HSP70s present in the host cytosol. This was carried out by fusing a known HsHSP70 substrate motif from HSF1 to the C-terminus of PFA66, providing both the J-domain protein and substate together so that HSP70-stimulation can be observed even when the J-domain itself is not a bona fide substrate. Using this system in single turn-over ATPase assays, a significant increase in activity for the parasite HSP70x as well as the two host HSP70s present in the RBC cytosol (HsHSP70 and HsHSC70) is observed over the substrate alone. Given that the PFA66 HPD motif is required to rescue the eKnob phenotype and that knockout of HSP70x (the only parasite HSP70 exported to the host cytosol) does not produce a knob phenotype, this result strengthens the implication that PFA66 performs its role through a partner HSP70 of host origin.

Reviewer #3: The revised manuscript addresses a key knowledge gap in how Plasmodium parasites create knob-like structures on the surface of the host RBC. The knobs act as antigen-presenting platforms and play a critical role in the pathogenesis of malaria. The data are excellent, experiments are well controlled, and the conclusions are mostly sound. The authors should be commended on discovering an interesting mechanism for Plasmodium knob formation, one I suspect will greatly help understand malaria pathogenesis.

The in vitro assays demonstrating the enhancement of human Hsp70’s APTase activity by PFA66 are very well done. In particular, the inclusion of the minimal substrate peptide shows that it is the tripartite complex that is essential for this activation. The assays are well controlled and add much to the discussion.

**Part II – Major Issues: Key Experiments Required for Acceptance**

Reviewer #1: I have already critiqued this paper for Review Commons where the major issue was a lack of physical evidence for the interaction of PFA66 with human and parasite HSP70s. Physical interactions of HSPs are highly transient and difficult to capture but the biochemical assays presented largely address the protein interaction issues. Other shortcomings with the PCR and western blot validations of the integration and expression of the GFP insertion into the PFA66 gene have been resolved.

Reviewer #2: (No Response)

Reviewer #3: (No Response)

**Part III – Minor Issues: Editorial and Data Presentation Modifications**

Reviewer #1: All the minor issues from the previous version appear to have been addressed.

Reviewer #2: Minor comments

-Line 122: Please state the expected fusion sizes for the skipped and unskipped as well as the PFA66-GFP fusion in the text and/or in the figure. I would also recommend providing a reference to the observation that these T2A fusions typically only skip ~50% of the time in Plasmodium, such as PMID 24160265 or 31164473.

-Line 911: Given that ~50% of the protein is not skipped, I would change “ensured” to something like “mediated”. Also, the * and + designations on the WB should be described in the figure or figure legend.

-Figure S1B: it appears the 43 kD marker is pointing to the top of the anti-GFP blot. I assume this should be the bottom of the Aldolase blot. Please adjust.

-The complementation line is introduced as ∆PFA[PFA::HA] in line 132 but then this name is not used in lines 185-187 when it is introduced again as a control for the eKnob phenotype. I assume it’s the same line – if so I would refer to it consistently to avoid any confusion.

-Were specific length or width thresholds use to quantify knob morphology categories in Figure 2C, 5D and S6B,D,E or was this done by eye? I didn’t find this in the methods or legends – sorry if I missed it but if it is not there, please add clarification on how the quantification was done.

-Line 192: “the lumen of these eKnobs was often extremely dense” It appears that it is not the lumen (or core) but the tip of the eKnob that is electron dense by TEM while the lumen of the eKnobs shown in Fig 2D, 3E and S3 is generally less dense. The tomography seems to show a dense core but I’m not seeing it in the TEMs being described here.

-Line 219-220: Were these other exported proteins quantified in the same way? The main text here states “no significant difference” which implies a statistical test was applied but the legend for Fig S4 says “no drastic difference”. From the response comments, it appears that the authors were unable to identify a difference by eye in a blinded check of the images. If so, “no significant difference” is misleading and should be removed from the text.

-Line 1005: The legend indicates GBP was detected in S1B but it is not there in the figure.

-Line 1023-1025: Please state the expected size of the KAHRP-mCherry fusion in the legend.

Line 326: references #43 and #44 are not right here. I think you mean Daniyan et al (#39) and Day et al 2019 (which appears to have been inadvertently deleted from the revised manuscript).

-Lines 364-366: The authors should note that some NPP phenotypes are obscured by rich media (for example, Gupta et al 2020, PMID: 32069335). The co-chaperone function of PFA66 could play a role in supporting the establishment of PSAC/NPPs at a level that does not produce a fitness cost without nutrient limitation. Seems this warrants a comment in the discussion.

Grammar

-Line 238: should be “imaged by STED microscopy”

-Line 469: “lead to misleading” is repetitive. Consider revising to “produce misleading”

-Line 951: “remaining” and “remained” is repetitive. Consider revising.

Reviewer #3: Lines 329-228: The JDS isn’t defined, what were the amino acid numbers?

Line 655: The ‘S’ in JDS is lowercase unlike in the rest of the manuscript.

Line 674: What was the source of ULP1 protease?

Line 714: There’s a typo, ‘JDH’ instead of ‘JDS’

PLOS authors have the option to publish the peer review history of their article (what does this mean?). If published, this will include your full peer review and any attached files.

Reviewer #1: No

Reviewer #2: **Yes: **Josh Beck

Reviewer #3: No

Figure Files:

Data Requirements:

Reproducibility:

References:

---

## [Editor Report · Decision Letter 2]

24 Sep 2021

Dear Dr Przyborski,

We are pleased to inform you that your manuscript 'Co-chaperone involvement in knob biogenesis implicates host-derived chaperones in malaria virulence' has been provisionally accepted for publication in PLOS Pathogens.

Best regards,

Kirk W. Deitsch

Section Editor

PLOS Pathogens

Kirk Deitsch

Section Editor

PLOS Pathogens

Kasturi Haldar

Editor-in-Chief

PLOS Pathogens

orcid.org/0000-0001-5065-158X

Michael Malim

Editor-in-Chief

PLOS Pathogens

orcid.org/0000-0002-7699-2064
---

## [Editor Report · Acceptance letter]

4 Oct 2021

Dear Dr Przyborski,

We are delighted to inform you that your manuscript, "Co-chaperone involvement in knob biogenesis implicates host-derived chaperones in malaria virulence," has been formally accepted for publication in PLOS Pathogens.

Best regards,

Kasturi Haldar

Editor-in-Chief

PLOS Pathogens

orcid.org/0000-0001-5065-158X

Michael Malim

Editor-in-Chief

PLOS Pathogens

orcid.org/0000-0002-7699-2064